

# Hydrological cycle amplification imposes spatial pattern on climate change response of ocean pH and carbonate chemistry

Allison Hogikyan [1,2] and Laure Resplandy [3,4]

[1]Atmospheric and Oceanic Sciences Program, Princeton University, Sayre Hall, 300 Forrestal Rd, Princeton, NJ 08540
[2]Department of the Geophysical Sciences, University of Chicago
[3]Geosciences Department, Princeton University
[4]High Meadows Environmental Institute, Princeton University

**Correspondence:** Allison Hogikyan (hogikyan@uchicago.edu)

**Abstract.** Ocean $CO_2$ uptake and acidification in response to human activities are driven primarily by the rise in atmospheric $CO_2$, but are also modulated by climate change. Existing work suggests that this 'climate effect' influences the uptake and storage of anthropogenic carbon and acidification via the global increase in ocean temperature, although some regional responses have been attributed to changes in circulation or biological activity. Here, we investigate spatial patterns in the climate effect on

surface-ocean acidification (and the closely related carbonate chemistry) in an Earth System Model under a rapid $CO_2$-increase scenario, and identify another culprit. We show that the amplification of the hydrological cycle, a robustly simulated feature of climate change, is largely responsible for the spatial patterns in this climate effect at the sea surface. This 'hydrological effect' can be understood as a subset of the total climate effect which includes warming, hydrological cycle amplification, circulation and biological changes. We demonstrate that it acts through two primary mechanisms: (i) directly diluting or concentrating dis-

solved ions by adding or removing freshwater and (ii) altering the sea surface temperature, which influences the solubility of dissolved inorganic carbon (DIC) and acidity of seawater. The hydrological effect opposes acidification in salinifying regions, most notably the subtropical Atlantic, and enhances acidification in freshening regions such as the western Pacific. Its single strongest effect is to dilute the negative ions that buffer the dissolution of $CO_2$, quantified as 'Alkalinity'. The local changes in Alkalinity, DIC, and pH linked to the pattern of hydrological cycle amplification are as strong as the (largely uniform) changes

due to warming, explaining the weak increase in pH and DIC seen in the climate effect in the subtropical Atlantic Ocean.

## 1 Introduction

The increasing atmospheric concentration of carbon dioxide ($CO_2$) causes a flux of $CO_2$ into the ocean, typically termed the '$CO_2$-concentration feedback' (here, we will use the term '$CO_2$ effect'). This oceanic $CO_2$ uptake increases the total carbon content of the ocean (total dissolved inorganic carbon; DIC), decreases the availability of buffering ions (Alkalinity

or Alk), and consequently leads to ocean acidification (decrease in pH: $-\log_{10}[H^+]$). This study links the enhancement of the hydrological cycle with warming to regional changes in DIC, Alkalinity, and acidification, thus linking a robust physical response of the climate system to a biological impact of climate change. Ocean acidification reduces the stability of solid calcium carbonate, weakening the protective shells of marine organisms, with negative impacts already visible, for example on





tropical coral reefs (e.g., Caldeira and Wickett, 2003; Gattuso et al., 2014). Acidification can also acts in synergy with other
stressors, including warming and ocean de-oxygenation, increasing the vulnerability of certain marine organisms. For example,
combined acidification and low oxygen levels narrow the range of temperatures at which organisms can function, and warming
tends to increase baseline metabolic rates, further narrowing this thermal window (e.g.,  ; Doney et al., 2020; Kroeker et al.,
2013).

Chemical changes in DIC, Alkalinity and pH are also modulated by climate change, via warming, circulation, freshwater
flux, and biological changes. These effects can be isolated from the direct $CO_2$ effect as a separate 'carbon-climate feedback'
(here we will use the term 'climate effect', including all changes other than the atmospheric $CO_2$ increase, i.e. temperature,
circulation, freshwater flux, and biological). This 'climate effect' has been shown to decrease global $CO_2$ uptake and storage
by approximately 10% in projections of high carbon emissions scenarios (Arora et al., 2013; Friedlingstein and Prentice, 2010;
Williams et al., 2019; McNeil et al., 2007; Schwinger et al., 2014). The best understood facet of the climate effect on seawater
chemistry is warming, which drives a decrease in anthropogenic carbon uptake due to weakened solubility and ventilation
(Katavouta and Williams, 2021). However, Katavouta and Williams (2021) also note that some regional patterns cannot be
accounted for by these two processes, and indeed other studies have suggested that more complex shifts in circulation patterns
and changes in biological activity contribute to regional climate effects on carbon storage in the interior (e.g. Lovenduski et
al., 2008; Siedlecki et al., 2021; Pilcher et al., 2019). Although these studies demonstrate that regional variations in the climate
effect are not exclusively related to warming, they are focused on anthropogenic carbon uptake and do not address the net
effect of DIC and Alkalinity changes on regional ocean acidification. McNeil et al. (2007) describe the climate effect on ocean
acidification and point out that warming has a direct effect to decrease pH and an indirect effect to decrease the solubility of
$CO_2$, which limits acidification. They find that the net of these direct and indirect effects is small on global average, but did not
address the larger regional responses, which are critical for anticipating ecosystem impacts. Here, we find that the amplification
of the hydrological cycle with warming, a robust response of climate models to global warming which has recently been linked
to patterns of ocean oxygen loss with climate change (Hogikyan et al., 2024), is in fact responsible for the bulk of the spatial
pattern in surface ocean DIC and Alkalinity and consequently acidification attributed to the climate effect.

The 'hydrological cycle amplification' reinforces sea surface salinity (SSS) patterns, leading to a 'salty-get-saltier, fresh-
get-fresher' rule of thumb for changes in SSS, especially at low and mid-latitudes where the hydrological cycle is strongest
(Durack and Wijffels, 2010, change in SSS associated with climate change in Figure 1a). Specifically, hydrological cycle
amplification refers to enhanced spatial patterns of net air-sea freshwater fluxes (precipitation-evaporation) which are largely
responsible for the mean SSS patterns (Held et al., 2006; Manabe and Wetherald, 1975). Hydrological cycle amplification can
induce regional patterns in surface seawater carbonate chemistry in two ways. First, freshwater fluxes can directly change the
concentration of dissolved species, potentially increasing the concentrations of DIC and Alkalinity in 'salty-get-saltier' regions
and decreasing their concentrations in 'fresh-get-fresher' regions. pH increases with DIC and decreases with Alkalinity, so that
freshwater fluxes drive a small net change in pH. Second, these changes in salinity modify the ocean circulation and lead to
enhanced ocean heat uptake (primarily in the subsurface subtropical Atlantic Ocean), which weakens surface warming (Liu et



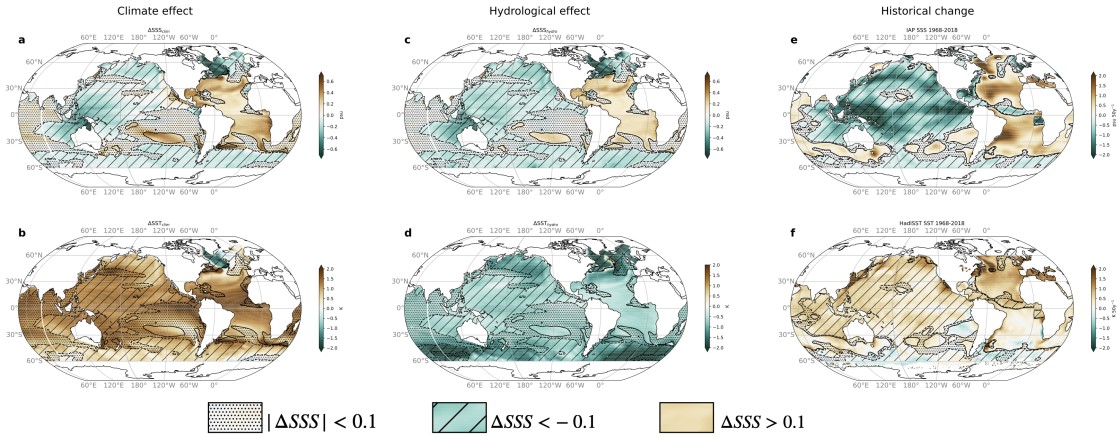

**Figure 1. Sea surface salinity (SSS) and temperature (SST) changes associated with climate and hydrological effects.** (a, b) SSS and SST change with climate effect (Standard minus Fix-Clim experiments) which includes hydrological effect and other changes, notably global warming; (c, d) SSS and SST change with hydrological effect (Standard minus Fix-SSS experiments) which weakens surface warming and reinforces SSS patterns. Also shown for comparison is the recent historical linear trend pattern in (e) SSS and (f) SST, as quantified by the IAP reanalysis (Cheng et al. , 2017) and the HadISST analysis product (Rayner et al. , 2003). Stippling indicates regions where $\Delta SSS_{hydro}$ is small ($|\Delta SSS_{hydro}| < 0.1$ psu); these regions are excluded from our analysis. Hatching indicates the fresh-get-fresher regime where $\Delta SSS_{hydro} < -0.1$ psu. A lack of hatching indicates the salty-get-saltier regime where $\Delta SSS_{hydro} > +0.1$ psu.

al., 2021; Williams et al., 2007, see also Figure 1b). This relative cooling could drive an increase in DIC and pH, weakening the influence of warming in the total climate effect.

We estimate the regional DIC, Alkalinity, and pH changes due to (a) the total climate effect and (b) the subset of the climate effect due only to hydrological cycle amplification- the 'hydrological effect'- using a high-$CO_2$-increase scenario in a global Earth System Model (NOAA-GFDL's ESM2M; (Dunne et al., 2013)). The experiments follow those used to isolate the hydrological effect on ocean oxygen loss in (Hogikyan et al., 2024). We focus on the sea surface, where the response to hydrological cycle amplification is largest, and separate the surface into a 'fresh-get-fresher' and a 'salty-get-saltier' regime; the

changes in these two regimes largely cancel in the global average but could modify local carbonate chemistry and its biological impacts. Then, we further attribute the climate effect and hydrological effect DIC, Alkalinity, and pH changes to two primary mechanisms: (a) freshwater (dilution and concentration of DIC and Alkalinity), and (b) thermal (temperature-driven) effects. We find that hydrological cycle amplification can account for much of the regional pattern in the total climate effect, since it is the sole driver of long-term trends in freshwater fluxes, while temperature changes are more spatially uniform. We show that

these freshwater fluxes change the concentration of Alkalinity slightly more than that of DIC due to the mean chemistry of the ocean, with the consequence that DIC, and pH tend to increase along with Alkalinity in 'salty-get-saltier' regions and decrease in 'fresh-get-fresher' regions.





## 2  Methods

### 2.1  Earth System Model and experiments to isolate hydrological and climate effects

We use the Geophysical Fluid Dynamics Laboratory Earth System Model 2M (ESM2M), which is fully described in (Dunne et al., 2012, 2013). The version of ESM2M used here (public release 5.0.2) uses the atmosphere model AM2 with a horizontal resolution of approximately 50 km, and ocean model MOM5 with a horizontal resolution of approximately 100km, as well as the land model LM3.0 and ocean biogeochemical model TOPAZ2. In order to isolate the signature of increasing atmospheric $CO_2$ and the associated amplification of the hydrological cycle, we force the model with a strong atmospheric $CO_2$ increase of

1% per year, beginning from the pre-industrial concentration of 286 ppm, until the $CO_2$ level doubles after 70 years. During years 71-100 the $CO_2$ level is held fixed at double the pre-industrial concentration (562 ppm), so that the entire experiment is 100 years long. We compare three different experiments with this 1%-to-doubling $CO_2$ forcing. This is identical to the experimental setup used in (Hogikyan et al., 2024), and further details can be found therein. The spatial patterns of salinity changes agree with long-term trends in observations and other climate models.

In the first experiment, the model freely responds to the prescribed atmospheric $CO_2$ ('Standard'), and the strength of the hydrological cycle intensifies with warming, amplifying patterns of freshwater fluxes and SSS. In the second experiment, SSS is nudged towards its pre-industrial monthly climatology with a restoring flux of freshwater ('Fix-SSS') as $CO_2$ increases along the same 1%-to-doubling trajectory. There is no restoring under seasonal sea ice. The freshwater restoring flux dilutes (or concentrates) all chemical species, although SSS is used to determine its strength. The difference between the Standard and Fix-SSS experiments provides an estimate of the impact of the hydrological cycle amplification on the ocean (including the direct freshwater flux and ocean circulation adjustment), and for a given variable X we define the change due to hydrological cycle amplification, the 'hydrological effect', in terms of the difference between these two simulations:

$$\Delta X_{hydro} = X_{Standard} - X_{Fix-SSS}$$

To contextualize the hydrological effect as a part of the total carbon-climate feedback, or 'climate effect', we also run a 'Fixed-Climate' experiment in which the same 1% to doubling atmospheric $CO_2$ increase interacts with ocean biogeochemistry but not with radiation so that there is no global warming (no change in climate). In this experiment, the ocean experiences carbon uptake due to atmospheric $CO_2$ increase, but no warming or hydrological cycle amplification. The climate effect can therefore be defined by

$$\Delta X_{clim} = X_{Standard} - X_{Fix-Clim}$$

(as in e.g. Williams et al., 2007, 2019; Katavouta and Williams, 2021). In this framework, we define $X_{Standard}$, $X_{Fix-Clim}$ and $X_{Fix-SSS}$ using the average over the last 30 years of each simulation (years 71-100) when atmospheric $CO_2$ is held steady at double the pre-industrial concentration and the system is beginning to equilibrate at this higher $CO_2$ level, to decrease the influence of internal variability or rapid adjustments to forcing. Results are presented for the Atlantic, Indian, and Pacific Oceans, with a focus on the low- and mid-latitudes where the hydrological cycle is most active, i.e. where the amplification

of evaporation/precipitation patterns is strongest. We ignore high latitudes where the surface freshwater balance is instead





dominated by ice-ocean interactions (north of 55°S and with a mask applied where seasonal sea ice is found in the pre-industrial control run; see mask in Figure 1).

$\Delta X_{clim}$ includes a contribution from $\Delta X_{hydro}$ as well as from other changes. For instance, the climate effect includes a significant surface warming, so that $\Delta SST_{clim}$ is positive (i.e., $SST_{Standard}$ exceeds $SST_{Fix-Clim}$ at the end of the simu-
lation,Figure 1c). However, the hydrological cycle amplification moderates surface warming by enhancing ocean heat uptake (Williams et al., 2007; Liu et al., 2021), so that $\Delta SST_{hydro}$ is negative (i.e., $SST_{Standard}$ is less than $SST_{Fix-SSS}$ at the end of the simulation, Figure 1d).

## 2.2   Freshwater and thermal contributions to DIC and Alkalinity changes

We quantify the influence of freshwater fluxes (dilution/concentration of DIC and Alkalinity) and temperature changes on DIC
and Alkalinity in both the hydrological effect (the amplification of the hydrological cycle, represented by the Standard - Fix-SSS experiments) and the climate effect (the total effect of climate change including warming, hydrological cycle amplification, etc., represented by the Standard - Fix-Clim experiments).

DIC is affected by both freshwater fluxes and temperature changes, so that we can decompose $\Delta DIC_{hydro}$ as follows:

$$\Delta DIC_{hydro} = \Delta DIC_{FW,hydro} + \Delta DIC_{thermal,hydro} + R_{DIC,hydro}$$

where $\Delta DIC_{FW,hydro}$ and $\Delta DIC_{thermal,hydro}$ correspond to contributions from dilution or concentration by freshwater fluxes (FW) and changes due to a temperature change (thermal). The residual $R$ includes all other processes that affect DIC
and Alkalinity (e.g., air-sea fluxes of $CO_2$, calcium carbonate precipitation/dissolution, production and remineralization of organic matter, and salinity) as well as co-variations between the thermal and freshwater effects and errors in our method of estimating these effects (which are elaborated on below).

In contrast, Alkalinity is not directly sensitive to temperature so that its hydrological effect can be approximated as:

$$\Delta Alk_{hydro} = \Delta Alk_{FW,hydro} + R_{Alk,hydro}$$

In analogy, we can attribute the total climate effect in DIC and Alkalinity to freshwater and thermal effects:

$$\Delta DIC_{clim} = \Delta DIC_{FW,clim} + \Delta DIC_{thermal,clim} + R_{DIC,clim}$$

$$\Delta Alk_{clim} = \Delta Alk_{FW,clim} + R_{Alk,clim}$$

The various residuals $R$ are quantified in Figures S1 and S2. We are largely successful in reconstructing the hydrological effect, and $R$ is generally small relative to $\Delta DIC_{hydro}$ and $\Delta Alk_{hydro}$ (error <5 $\mu$mol/kg at a point relative to broad re-
gional changes of 15-40 $\mu$mol/kg; Figures 2, A1). The error is somewhat more significant in reconstructing the climate effect ($\Delta DIC_{clim}$ and $\Delta Alk_{clim}$, Figure A2), especially for DIC (error <8 $\mu$mol/kg). This is most likely due to the fact that the climate effect leads to anomalous air-sea $CO_2$ fluxes (primarily due to warming, but possibly also influenced by circulation changes) which change DIC and can indirectly lead to changes in Alkalinity.





We quantify the effect of freshwater fluxes on Alkalinity and DIC ($\Delta\text{Alk}_{FW,hydro}$, $\Delta\text{DIC}_{FW,hydro}$) with a simple conserva-
tion argument which neglects the effects of mixing and advection, a fair approximation within the mixed layer. In this case, DIC,
Alkalinity, and salt are diluted/concentrated by air-sea freshwater fluxes by the same fraction $f_{FW,hydro} = \Delta\text{S}_{hydro}/\text{S}_{Fix-SSS}$
referenced to salinity (S). For example, if a freshwater flux into the surface in the Standard warming experiment diluted SSS
by $f_{FW,hydro} = 5\%$ compared to the Fix-SSS experiment (i.e., $\text{SSS}_{Standard} = 0.95\,\text{SSS}_{Fix-SSS}$), then surface DIC and Alka-
linity would also be diluted by $f_{FW,hydro} = 5\%$ from the reference Fix-SSS concentrations ($\text{DIC}_{Standard} = 0.95\,\text{DIC}_{Fix-SSS}$;
$\text{Alk}_{Standard} = 0.95\,\text{Alk}_{Fix-SSS}$). We can therefore approximate $\Delta\text{DIC}_{FW,hydro}$ and $\Delta\text{Alk}_{FW,hydro}$ as:

$$
\begin{aligned}
\Delta\text{Alk}_{FW,hydro} &= f_{FW,hydro}\,\text{Alk}_{Fix-SSS}, \\
\Delta\text{DIC}_{FW,hydro} &= f_{FW,hydro}\,\text{DIC}_{Fix-SSS}
\end{aligned}
\tag{1}
$$

This approach is similar to the standard salinity normalization (Broecker and Peng, 1992; Robbins, 2001). However, we ref-
erence DIC and Alkalinity to the scenario with no hydrological cycle amplification (Fix-SSS experiment) whereas work using
salinity normalization used a central salinity of 35 psu and DIC and Alkalinity values that tend to co-occur with this salinity
of 35 psu. This approach includes the influences of mixing and transport (since they are included in the changes in salinity
used to define $f_{FW}$) but assumes they affect salinity, Alkalinity, and DIC equally. Mixing and transport act on different spa-
tial gradients for each variable and would therefore lead to different concentration changes, so we expect that this approach
will work best in the mixed layer where all quantities are well mixed. As a result, we focus on explaining changes in surface
concentrations, and we find that it successfully reconstructs changes in surface concentrations.

We estimate DIC changes due to thermal changes following (Sarmiento and Gruber, 2006), using a constant thermal sensi-
tivity $\frac{\partial DIC}{\partial T}$ of -7 $\frac{\mu mol/kg}{K}$:

$$
\Delta\text{DIC}_{thermal,hydro} = \frac{\partial DIC}{\partial T}\Delta\text{T}_{hydro} = -7\Delta\text{T}_{hydro}
$$

This approximation introduces some error since $\frac{\partial DIC}{\partial T}$ is not constant, but we find that our results do not change if we allow
the sensitivity to vary at each model grid point and month. Our conclusions are not sensitive to the choice of constant within a
range of $7\pm2\ \frac{\mu mol/kg}{K}$.

## 2.3 Attribution of surface pH changes to hydrological and climate effects

Equilibrium pH can be understood as a nonlinear function of DIC, Alkalinity, temperature (T), and salinity (S), so that a differ-
ence in pH between two model experiments or ocean chemical states can be interpreted in terms of the corresponding changes
in DIC, Alkalinity, temperature, and salinity between these two states (as in e.g. García-Ibáñez et al., 2016). An increase in DIC
due to $CO_2$ dissolution produces $H^+$ ions and decreases pH, whereas an increase in Alkalinity represents a greater seawater
buffering capacity and yields a higher pH. Temperature has a direct negative relationship with pH (warming ionizes water, thus
decreasing pH), and an indirect positive relationship with pH (the solubility of DIC decreases with temperature and leads to an
increase in pH). Salinity has a small effect on pH and will not be discussed in this study.





We define the 'hydrological effect' and 'climate effect' on pH as:

$$\Delta pH_{hydro} = pH(DIC_{Std}, Alk_{Std}, T_{Std}, S_{Std}) - pH(DIC_{Fix-SSS}, Alk_{Fix-SSS}, T_{Fix-SSS}, S_{Fix-SSS})$$ (2)

$$\Delta pH_{Clim} = pH(DIC_{Std}, Alk_{Std}, T_{Std}, S_{Std}) - pH(DIC_{Fix-Clim}, Alk_{Fix-Clim}, T_{Fix-Clim}, S_{Fix-Clim})$$ (3)

using the marine carbonate chemistry solver PyCO2SYS (Humphreys et al, 2022). pH is a highly non-linear function of other state variables, and is solved for iteratively. We therefore make use of this established solver rather than making our own estimate, as we do for DIC and Alk. These $\Delta pH_{Clim}$ and $\Delta pH_{Hydro}$ estimates are not identical to $pH_{Std} - pH_{Fix-Clim}$ and $pH_{Std} - pH_{Fix-Hydro}$ since CO2SYS assumes chemical equilibrium while ESM2M does not (it is constrained instead by the conservation of heat and mass in the coupled model). We use this method because it allows us to break down $\Delta pH_{hydro}$ and $\Delta pH_{Clim}$ into freshwater (chemical dilution) and thermal (temperature-driven) components (as well as a residual due to errors in method and other drivers). The freshwater effect is defined:

$$\Delta pH_{hydro} = \Delta pH_{FW,hydro} + \Delta pH_{thermal,hydro} + R$$

Freshwater fluxes affect pH primarily through their effect on DIC and Alkalinity; we use the DIC and Alkalinity changes due to freshwater fluxes (as estimated in section 2.2) to evaluate the freshwater flux effect on pH. $\Delta pH_{FW,hydro}$ is defined as the difference between the theoretical pH with the DIC and Alkalinity predicted from the effect of freshwater fluxes (in bold) and and the reference $pH_{Fix-SSS}$ which excludes the influence of hydrological cycle amplification:

$$\Delta pH_{FW,hydro} = pH((\mathbf{f_{FW,hydro}} + \mathbf{1})\mathbf{DIC_{Fix-SSS}}, (\mathbf{f_{FW,hydro}} + \mathbf{1})\mathbf{Alk_{Fix-SSS}}, SST_{Fix-SSS}, SSS_{Fix-SSS}) - pH_{Fix-SSS}$$

Similarly, the net change in pH due to direct and indirect (via DIC) thermal effects is estimated as:

$$\Delta pH_{thermal,hydro} = pH(\mathbf{DIC_{Fix-SSS}} + \mathbf{\Delta DIC_{T,hydro}}, Alk_{Fix-SSS}, \mathbf{SST_{Standard}}, SSS_{Fix-SSS}) - pH_{Fix-SSS}$$

## 3 Results

### 3.1 Climate-driven DIC and Alkalinity changes explained by hydrological effect

We first compare the mean climate and hydrological effect averaged over ocean surface areas that experience salinification in excess of 0.1 psu (the (sub)tropical Atlantic and southeast Pacific Oceans) and those that experience freshening stronger than -0.1 psu (the high-latitude Atlantic and remainder of the Indo-Pacific Oceans, Figures 2, 1), to characterize the response of the surface ocean to freshening and salinification. Although the DIC change has a similar spatial pattern as Alkalinity, dictated by the sign of freshwater fluxes, we will show that its magnitude is modulated by a thermal component (warming in the climate effect and cooling in the hydrological effect). Where SSS increases, DIC is less sensitive than Alk to the climate





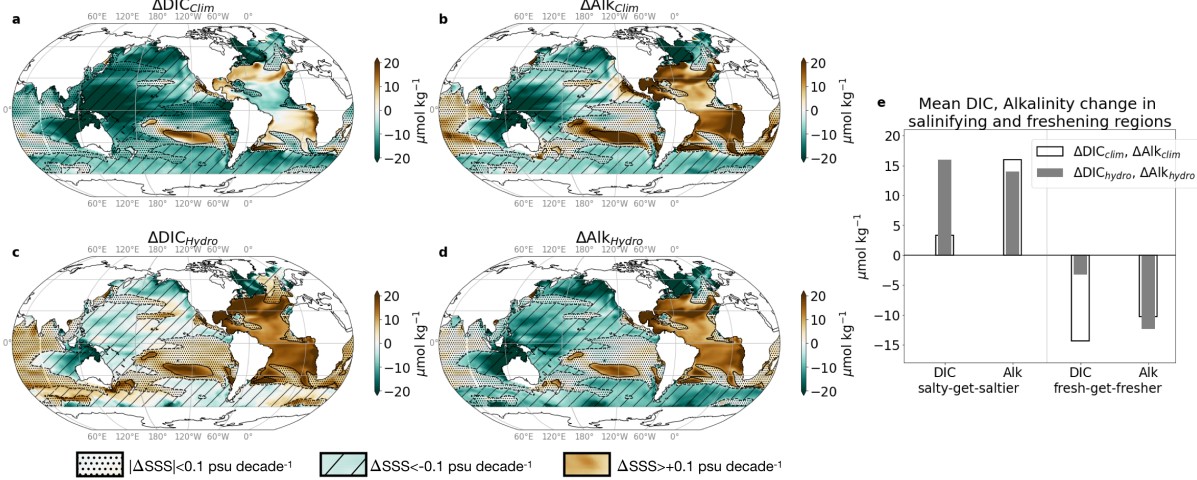

**Figure 2. DIC and Alkalinity response to climate and hydrological effects.** Change in surface (a) DIC and (b) Alkalinity due to the climate effect (Standard minus Fix-Clim experiments); change in surface (c) DIC and (d) Alkalinity due to the hydrological effect (Standard minus Fix-SSS experiments). (e) Mean change in DIC (black) and Alkalinity (grey) in salinifying ($\Delta SSS_{hydro} > 0.1$ psu) and freshening ($\Delta SSS_{hydro} < $ -0.1 psu) regions shown in Figure 1. For a-d, stippled areas experience nearly zero change in salinity ($|\Delta SSS_{hydro}| < 0.1$ psu) and are ignored in our analysis, while fresh-get-fresher regions are hatched ($\Delta SSS_{hydro} < $ -0.1 psu), and salty-get-saltier regions have no hatching ($\Delta SSS_{hydro} > $ +0.1 psu).

effect ( $\Delta DIC_{clim}$ = +2 $\mu$mol/kg and $\Delta Alk_{clim}$ = +16 $\mu$mol/kg), but they have the same sensitivity to the hydrological effect ($\Delta DIC_{hydro}$ = +16 umol/kg and $\Delta Alk_{hydro}$ = +14 umol/kg). Where SSS decreases, the two have a similar response to the climate effect ($\Delta DIC_{clim}$ = -13 umol/kg while $\Delta Alk_{clim}$ = -10 umol/kg, black empty bars in Figure 2), but DIC is less sensitive than Alk to the hydrological effect ($\Delta DIC_{hydro}$ = -5 umol/kg and $\Delta Alk_{hydro}$ = -12 umol/kg, grey bars in Figures 2e and 3). We
can understand what controls these changes in DIC and Alkalinity by attributing them to freshwater (dilution/concentration), thermal, and residual (e.g. biological or circulation) effects using simple sensitivity estimates described in the Methods section.

The effect of freshwater fluxes is very similar in both the hydrological effect and the climate effect, consistent with the understanding that hydrological cycle amplification accounts for the bulk of salinity changes in the total climate effect (Durack et al., 2012) (Figure 1a, c; i.e. $f_{FW,hydro} \approx f_{FW,clim}$). However, the higher mean concentration of Alkalinity makes Alkalinity
more sensitive to freshwater addition and removal than DIC (in both the total climate effect and hydrological effect; blue bars in Figure 3, maps in Figure A3). For example, when we evaluate ($\Delta X_{FW,hydro} = X_{Fix-SSS} f_{FW,hydro}$), $\Delta Alk_{FW,hydro}$ is slightly greater than $\Delta DIC_{FW,hydro}$ due to the greater reference $Alk_{Fix-SSS}$ (Figure 3 blue bars). The freshwater effect in salty-get-saltier waters increases DIC and Alkalinity by 17 and 19 umol/kg in the climate effect; the increase is slightly less in the hydrological effect, 12 and 14 umol/kg. Dilution from freshening decreases DIC and Alkalinity decrease -9 and -10
umol/kg in the climate effect. The freshwater effect on both DIC and Alk is slightly larger in the hydrological effect alone, -13 and -14 umol/kg. These discrepancies between the hydrological and climate effects are primarily due to the discrepancy





between dilution/concentration fractions $f_{FW,hydro}$ and $f_{FW,clim}$, which arise from the difference in ocean circulation and salinity fields between the Fix-SSS and Fix-Clim experiments.

Changes in surface temperature substantially modifies the DIC response to both the total climate effect and the hydro-
logical effect since its solubility decrease with temperature. Alkalinity, however, is not very sensitive to temperature and as a result its changes are well accounted for by the hydrological effect alone (yellow residual bars for Alk are much smaller than freshwater and total Alk changes; Figure 3). Surface warming in the climate effect decreases DIC concentrations every-where (by -9 umol/kg in salinifying and -7 umol/kg in freshening regions), while surface cooling in the hydrological effect increases DIC concentrations everywhere (by +5 umol/kg in salinifying and +7 umol/kg in freshening regions; Figure 3 green
bars). Overall, the SST and corresponding DIC changes are similar across salinifying and freshening regions (Figures 1c, d and S3b, e), although circulation changes lead to some spatial patterns in the temperature response. For example, the weaker $\Delta DIC_{thermal,clim}$ in freshening, relative to salinifying, regions (-6 vs. -8 umol/kg; Figure 3) is a consequence of the SST decrease (and positive $\Delta DIC_{thermal,clim}$) in the Labrador Sea associated with a weakening of the overturning circulation, a common transient response of climate models to global warming and North Atlantic freshening (Figure 1) (Menary and Wood,
2018; Manabe and Stouffer, 1995). Similarly, the greater $\Delta DIC_{thermal,hydro}$ in freshening regions (+7 vs. +5 umol/kg; Figure 3) is a consequence of slightly stronger cooling at high latitudes where deep isopycnal mixing enhances the surface tempera-ture response, while the cooling in the salty (sub)tropics is weaker. Despite these small differences, the $\Delta DIC_{thermal,hydro}$ and $\Delta DIC_{thermal,clim}$ are uniform in sign and quite similar in magnitude in both regimes, leading to a contrast between the net climate and hydrological effects where $\Delta DIC_{clim} < \Delta Alk_{clim}$, while $\Delta DIC_{hydro} > \Delta Alk_{hydro}$ (black empty and filled bars
in Figure 3). In summary, the sign and spatial pattern of DIC and Alkalinity changes in both the climate effect and hydrological effect are determined by freshwater fluxes associated with hydrological cycle amplification, and the magnitude of DIC changes is further modulated by changes in SST.

This simple decomposition into freshwater and thermal effects leaves some of the simulated changes in DIC and Alkalin-ity unexplained (with the residual error represented by the yellow bars in Figure 3). Major processes that are not included
in these freshwater and thermal effects include air-sea $CO_2$ fluxes, spatial shifts in the atmospheric and oceanic circulations, and biological activity. Despite these omissions, the decomposition skillfully reconstructs the hydrological effect in both DIC and Alkalinity ($R_{hydro} <2$ umol/kg for both; see also Figure A4), while residuals for the climate effect are slightly larger ($R_{clim} \approx$ 4-5 umol/kg for Alk and DIC; see also Figure A4). Biases in Alkalinity reconstruction are $<10\%$ of the total hy-drological effect, suggesting that this simple estimate of dilution is a fairly effective estimate of the influence of hydrological
cycle amplification on Alkalinity. The residuals in DIC are broadly consistent with the influence of air-sea $CO_2$ fluxes. For instance, the decomposition of the hydrological effect for DIC is biased slightly high in salty-get-saltier regions, consistent with unaccounted-for $CO_2$ outgassing (a secondary effect of the DIC increase and thus $pCO_2$ increase), and the inverse is true in fresh-get-fresher regions (Figure A5). Similarly, the negative bias (missing carbon) in the climate effect reconstruction in salinifying regions is consistent with anomalous $CO_2$ uptake linked to the decrease in DIC (Figure A5). We were not able
to develop a simple attribution of surface DIC changes to air-sea fluxes (as we can with freshwater fluxes and temperature changes) because the DIC change due to a given surface flux is sensitive to multiple factors, including the efficiency of 3D



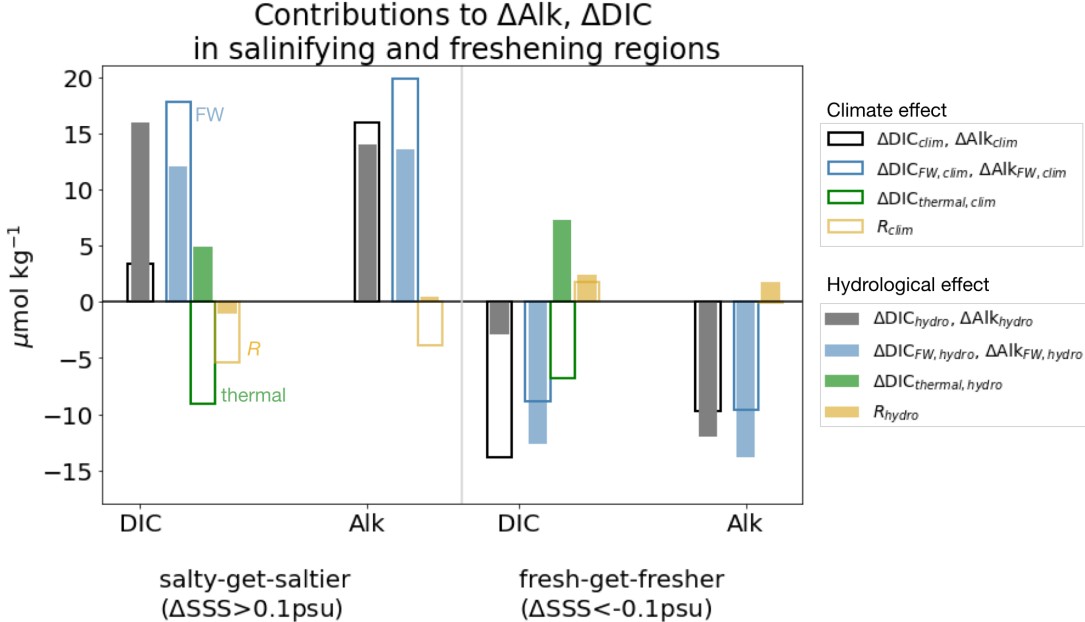

**Figure 3. Thermal and freshwater components of DIC and Alkalinity changes in 'fresh-get-fresher' and 'salty-get-saltier' regions.** Change in surface DIC and Alkalinity concentrations in response to the the climate effect (empty bars) and the hydrological effect (filled bars) in salinifying and freshening regions (as in Figures 4c and 2e). Black and grey bars represent total $\Delta_{clim}$ and $\Delta_{hydro}$, and are identical to those in Figure 2e. Blue bars represent change in DIC or Alkalinity due to freshwater fluxes. Green bars represent change in DIC due to SST change.

mixing in the near-surface ocean, which determines the local DIC and pCO$_2$ change, as well as the temperature, surface wind speed, and sea state.

## 3.2   Acidification weakened in 'salty-get-saltier' regions and exacerbated in 'fresh-get-fresher' regions

The climate effect tends to decrease surface pH, thereby re-inforcing the acidification associated with the rise in atmospheric CO2 ($\Delta$pH$_{clim}$ <0; Figure 4a). We find, however, that on average, the climate effect enhances acidification more in 'fresh-get-fresher' regions than in 'salty-get-saltier' regions (-0.005 vs. -0.001; black empty bars in Figure 4c). The hydrological effect (dilution/concentration of DIC/Alk, direct and indirect effect of SST decrease) contributes strongly to the changes in pH simulated in response to climate change, and largely explains the contrast in magnitude between freshening and salinifying regions (Figure 4). In particular, the hydrological effect contributes to the acidification in 'fresh-get-fresher' regions such as subpolar oceans, but opposes acidification in 'salty-get-saltier' regions such as the subtropical Atlantic (-0.001 vs. +0.002 on average over freshening and salinifying regions; grey bars in Figure 4c). Note that the climate-driven increase in pH simulated



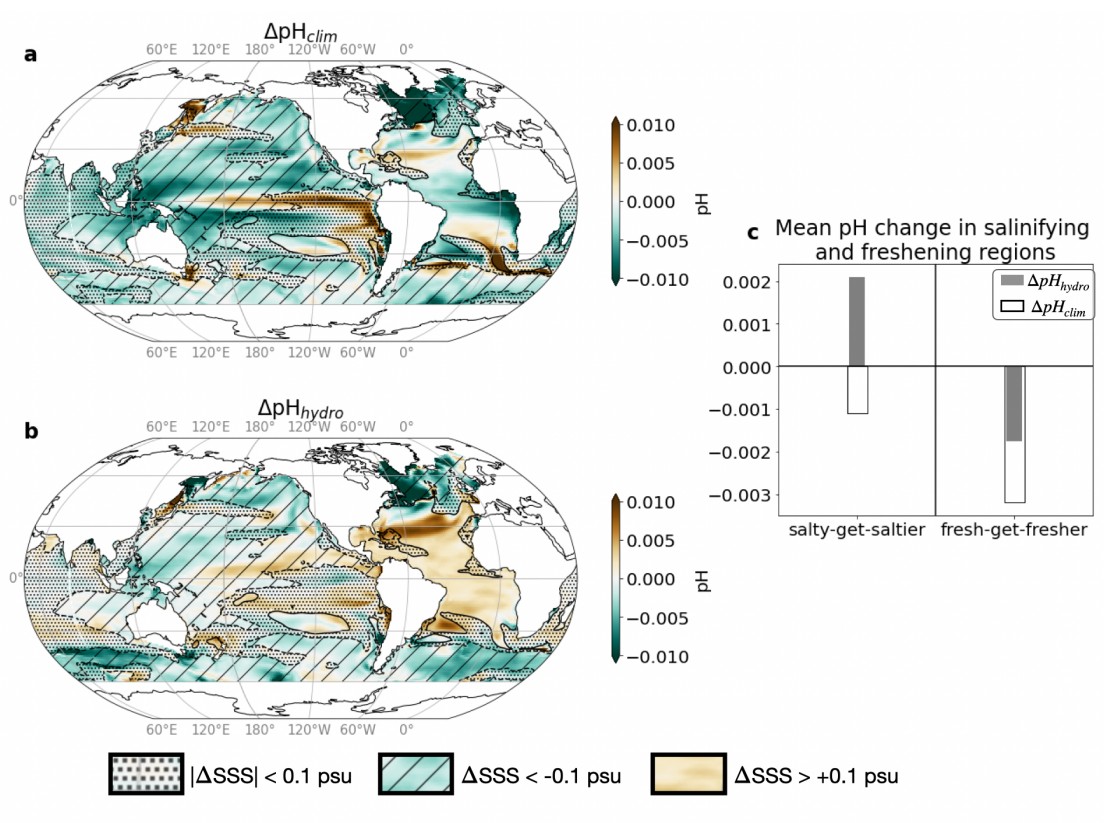

**Figure 4. pH response to climate and hydrological effects.** (a) Change in surface pH due to the climate effect (Standard minus Fix-Clim experiments). (b) Mean change in salinifying ($\Delta SSS_{hydro} > 0.1$ psu) and freshening ($\Delta SSS_{hydro} < $ -0.1 psu) regions outlined in Figure 1. Black contours indicate $\Delta SSS_{hydro}$ = +/-0.1 psu and hatching indicates $\Delta SSS_{hydro} < 0$. Stippled areas indicate regions that are neither salinifying nor freshening significantly ($|\Delta SSS_{hydro}| < 0.1$ psu) and are not used in the analysis. Panel (c) shows the mean pH change in salinifying (left) or freshening (right) regions, with empty bars representing the climate effect and solid bars representing the hydrological effect.




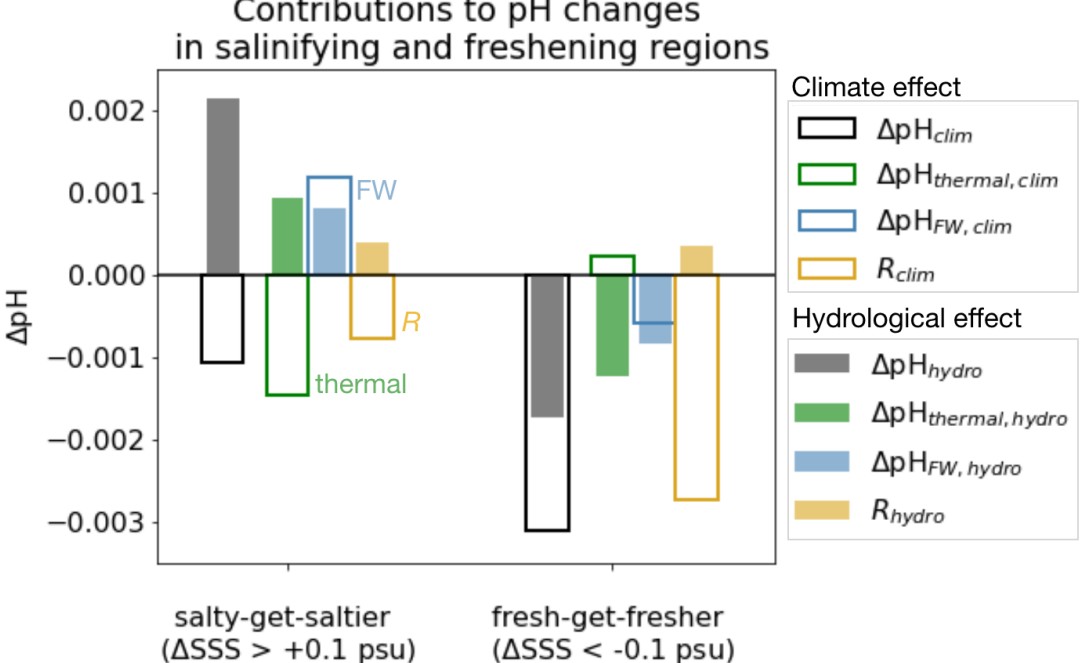

**Figure 5. Thermal and freshwater components of pH changes in 'fresh-get-fresher' and 'salty-get-saltier' regions.** Change in surface pH in the climate effect (empty bars) and the hydrological effect (filled bars), in 'salty-get-saltier' and 'fresh-get-fresher' regions (as shown in Figure 1). Black and grey bars represent total simulated pH change and are identical to those in Figure 4c. Blue bars represent contribution of freshwater effect (via dilution of DIC and Alkalinity). Green bars represent contribution of thermal effect (via SST change and DIC change due to SST). Yellow bars represent the residual (the difference between the sum of green + blue, and the actual change represented by black and grey bars).

in the equatorial Pacific upwelling region cannot be accounted for by the hydrological effect (Figure 4a-b), but is instead associated with the weakened upwelling of cold and high-DIC waters simulated in this region(Figures 1b and 2a).

We next interpret these pH changes due to the climate and hydrological effects in terms of a thermal effect (which includes the opposing effects of DIC and SST on pH) and a freshwater flux effect (which includes the opposing effects of DIC and Alkalinity on pH), using the thermal and freshwater components of DIC and Alkalinity changes presented in Section 3 (Figure 5).

The weakened acidification tied to the hydrological cycle in salinifying waters (i.e. an increase in pH tied to the hydrological

effect) is attributed almost equally to the freshwater and thermal effects (filled bars in Figure 5). pH has a similar sensitivity to both Alkalinity and DIC in the modern surface ocean, and $\Delta\mathrm{Alk}_{FW}$ always exceeds $\Delta\mathrm{DIC}_{FW}$ (since the FW contribution scales with the mean value, and the mean Alkalinity concentration is higher than the mean DIC concentration; see Methods). As as result, the increasing $\Delta\mathrm{Alk}_{FW}$ drives a positive $\Delta\mathrm{pH}_{FW}$ in salty-get-saltier waters (Figure 6c). At the same time,



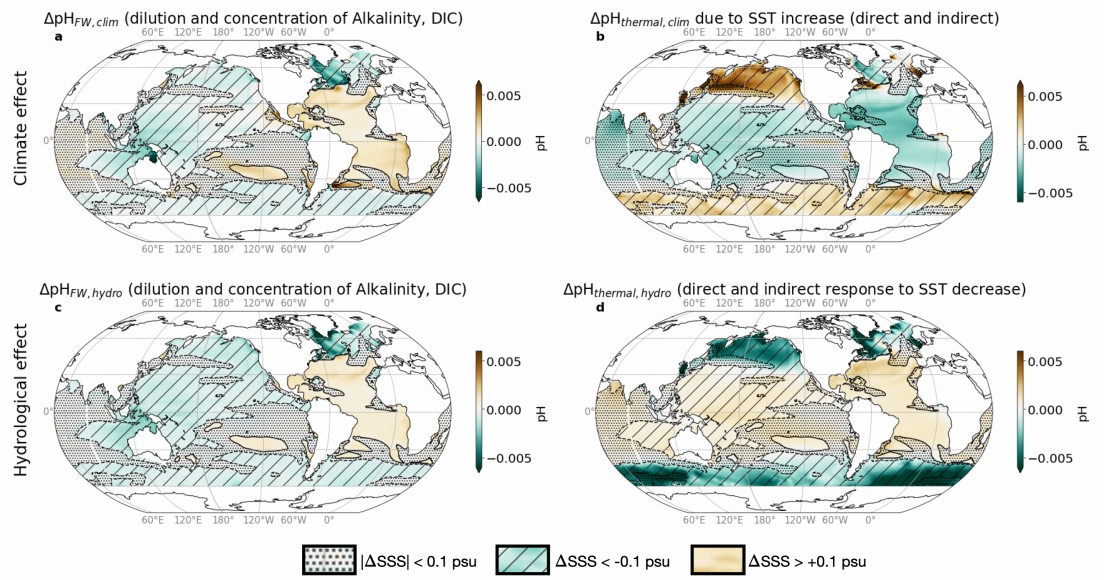

**Figure 6. Contributions of temperature and freshwater flux changes to climate and hydrological effect.** Upper row shows the contributions of (a) freshwater fluxes and (b) surface temperature change (warming) to the climate effect. Lower row shows the contributions of (c) freshwater fluxes and (d) surface temperature change (less warming) to the climate effect. Black contours indicate $\Delta SSS_{hydro}$ = +/-0.1 psu and hatching indicates $\Delta SSS_{hydro} < 0$.

hydrological cycle amplification drives a weak surface cooling (Liu et al., 2021) which further increases pH (McNeil et al.,
2007). At lower latitudes where salty-get-saltier waters are found, the direct effect of cooling (cooling increases pH) overcomes
the indirect effect (cooling increases DIC and reduces pH). Together, the increased Alkalinity and decreased temperature
increase pH, weakening the net acidification in the total climate effect. Similarly, the enhanced acidification in freshening
waters is also attributed almost equally to the freshwater and thermal effects (filled bars in Figure 5). The decrease in alkalinity
due to freshwater input reinforces acidification. In this case, the indirect effect of cooling on pH exceeds the direct effect
leading additional to acidification, in particular in cold, DIC-rich high latitude waters, where the fresh-get-fresher signal is
strongest (Figure 6 d).

This decomposition into freshwater and thermal effects is insightful but simplistic. The combined freshwater and thermal
effects correctly predict the sign of pH changes, but better reconstruct the magnitude of the hydrological effect than of the
climate effect (yellow residuals greater for climate effect in Figure 5). The sum of these two components underestimates
the magnitude of the total simulated pH decrease in the climate effect (predicting a weak decrease relative to the simulated
$\Delta pH_{clim}$ -0.001 and -0.003 in salinifying and freshening regions). The weakened air-sea $CO_2$ flux cannot explain the enhanced
acidification, suggesting that this inaccuracy is related to error in our reconstructions of $\Delta Alk_{FW,clim}$, $\Delta DIC_{FW,clim}$, and
$\Delta DIC_{thermal,clim}$, which are discussed above, as well as the covariation between drivers of pH changes which reflects the
non-linearity of the carbonate chemistry system.



Despite these limitations, the hydrological effect appears to contribute strongly to the spatial pattern of pH changes. The discrepancy in pH changes between the salty-get-saltier and fresh-get-fresher regions due to hydrological cycle amplification (nearly 0.004) is in fact larger than the discrepancy in the total climate effect (0.002). The hydrological effect introduces this strong pattern by (i) cooling the surface and (ii) freshwater dilution in freshening regions and concentration in salinifying regions.

## 4 Discussion and conclusions

Our results suggest that the changes in Alkalinity, DIC, and pH linked to hydrological cycle amplification contribute as strongly to the spatial pattern of the climate effect as warming alone, although these effects largely offset one another in the global mean. In fact, since Alkalinity is not strongly influenced by temperature, nearly the entire climate effect in surface Alkalinity is accounted for by the hydrological effect, i.e. dilution or concentration by anomalous freshwater fluxes (precipitation - evapo-

ration). The climate effect includes a surprising DIC and pH increase (opposite the decreases expected from surface warming) in the subtropical Atlantic Ocean, as well as an exceptionally strong DIC decrease at higher latitudes. Both of these features can be largely accounted for with the hydrological effect. Although the freshwater loss from the ('salty-get-saltier') subtropical Atlantic Ocean leads to an increase in both Alkalinity and DIC, the increase in Alkalinity is greater (and the DIC increase is weakened by the global cooling effect of hydrological cycle amplification), leading to a local increase in pH. The critical role

of Alkalinity in determining the response of marine carbonate chemistry to climate change here is consistent with prior studies (e.g. Chikamoto et al., 2023; Planchat et al., 2024). At high latitudes, the decrease in DIC associated with warming in the total climate effect is amplified by dilution ('fresh-get-fresher'), and tends to support stronger acidification. The nonlinear response of pH and other biologically important parameters (such as aragonite saturation state (e.g. Pinsonneault et al., 2012)) are left to other studies to study in more detail.

Although this is the first study to isolate the hydrological effect on seawater carbonate chemistry, the amplification of the hydrological cycle itself, and its impact on ocean heat uptake and SST, have been shown to be consistent across ocean-atmosphere models, despite differences in core ocean and atmosphere components (Williams et al., 2007; Liu et al., 2021; Held et al., 2006). Because of these prior results, we expect the sign and magnitude of these results to remain similar across other models. This study attempts to explain the mechanisms driving patterns in the total climate effect for DIC, Alkalinity, and pH,

and is intended to complement two recent studies of the influence of the hydrological effect on ocean heat uptake and ocean oxygen content and distribution (Liu et al., 2021; Hogikyan et al., 2024). We find regional changes due to the hydrological effect alone are the same order of magnitude as the total climate effect and can explain the spatial patterning in the climate effect (+/- 0.01 pH, +/-20$\mu$mol kg$^{-1}$ DIC, and +/-20 $\mu$mol kg$^{-1}$ Alk), but both are of course smaller than the direct effect of $CO_2$ (in these simulations, approximately -0.2 pH, 120 $\mu$mol kg$^{-1}$ DIC, and 20 $\mu$mol kg$^{-1}$ Alk) (Williams et al., 2019).

However, it is encouraging that we are able to clearly quantify this effect of freshwater fluxes even under relatively strong atmospheric $CO_2$ forcing, despite the strong non-linearity of ocean carbonate chemistry.





The surface patterns in DIC and Alkalinity presented here affect the distribution of carbon storage within the ocean. For example, the hydrological cycle leads to a slight increase in carbon storage in Mode and Intermediate Waters at the expense of deeper water masses, while the total climate effect tends to decrease carbon uptake and storage everywhere. However, these changes (due to both the hydrological effect and the total climate effect) are small in the ocean interior (generally $< |5|$ $\mu$mol kg$^{-1}$).

Finally, it is interesting to recall that the dilution and concentration of dissolved species by freshwater fluxes (our 'freshwater component') is traditionally removed by a salinity normalization (following Broecker and Peng, 1992). The prevalence of this practice implies that there is a strong influence of freshwater fluxes on seawater chemistry, but this influence is not commonly studied. We have demonstrated (see Methods) that, although the traditional salinity normalization is largely successful at removing the direct effect of freshwater fluxes on DIC and Alkalinity individually, but because Alkalinity has a greater mean value it will change more, leading to a (regionally) non-zero net change in pH, and potentially carbon uptake, due to freshwater fluxes. In some regions, the net effect (imbalance between Alkalinity and DIC) can be substantial, e.g. in North Atlantic Mode Waters where salinity increases strongly and Antarctic Intermediate Waters where it decreases strongly, highlighting the need to carefully consider if this salinity normalization is adequate and the influence of freshwater fluxes can be ignored.

*Code and data availability.* Processed model results and code to reproduce figures will be made available on Dataspace, and can be made available to reviewers upon request.

**Appendix A**

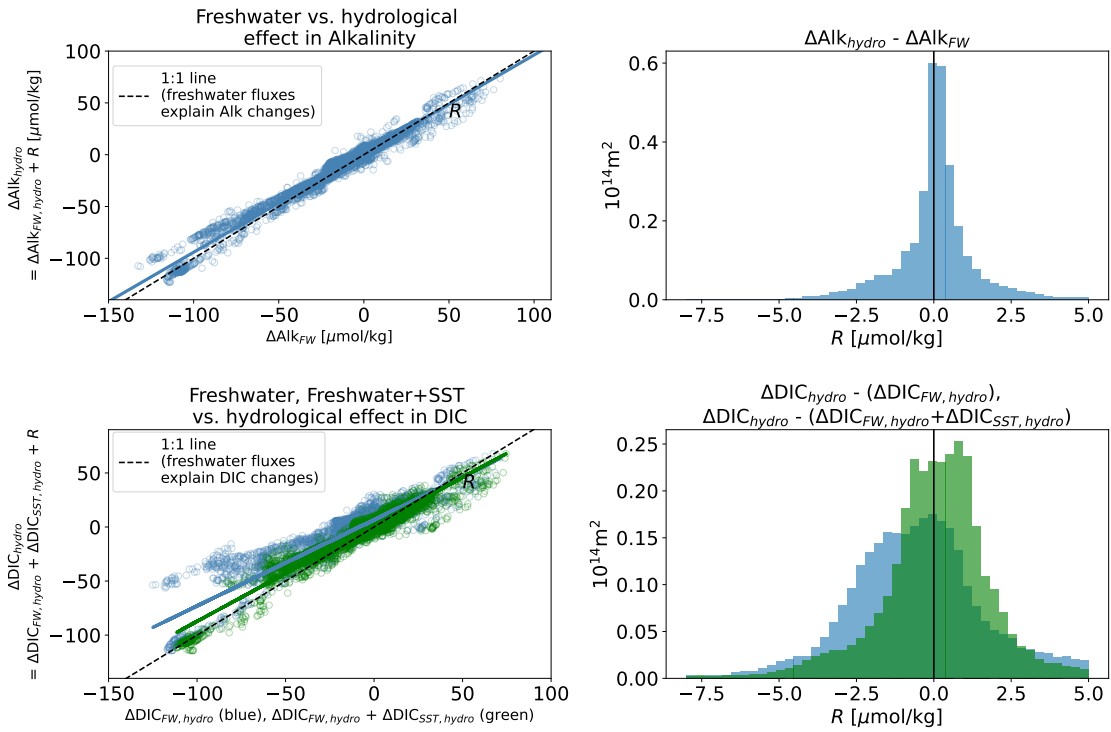

**Figure A1. Ability of freshwater, SST decomposition to reconstruct spatial pattern of changes in surface DIC and Alkalinity with hydrological effect.** Success of 'freshwater effect' ($\Delta\text{Alk}_{FW,hydro}$) in predicting $\Delta\text{Alk}_{hydro}$ (top), and success of 'freshwater + SST effects' ($\Delta\text{DIC}_{FW,hydro}+\Delta\text{DIC}_{SST,hydro}$) in predicting $\Delta\text{DIC}_{hydro}$ (bottom). Left hand side: scatter and unweighted regression of simulated surface $\Delta\text{Alk}_{hydro}$ or $\Delta\text{DIC}_{hydro}$ against the $\Delta_{hydro}$ predicted by our decomposition (for Alkalinity, $\Delta\text{Alk}_{FW,hydro}$; for DIC, $\Delta\text{DIC}_{FW,hydro}$ in blue and $\Delta\text{DIC}_{FW,hydro}+\Delta\text{DIC}_{SST,hydro}$ in green), at each surface location. Scatter around the fit is the error $R$. Right hand distributions represent area weighted values of $R$ for Alkalinity (top) and DIC (bottom).

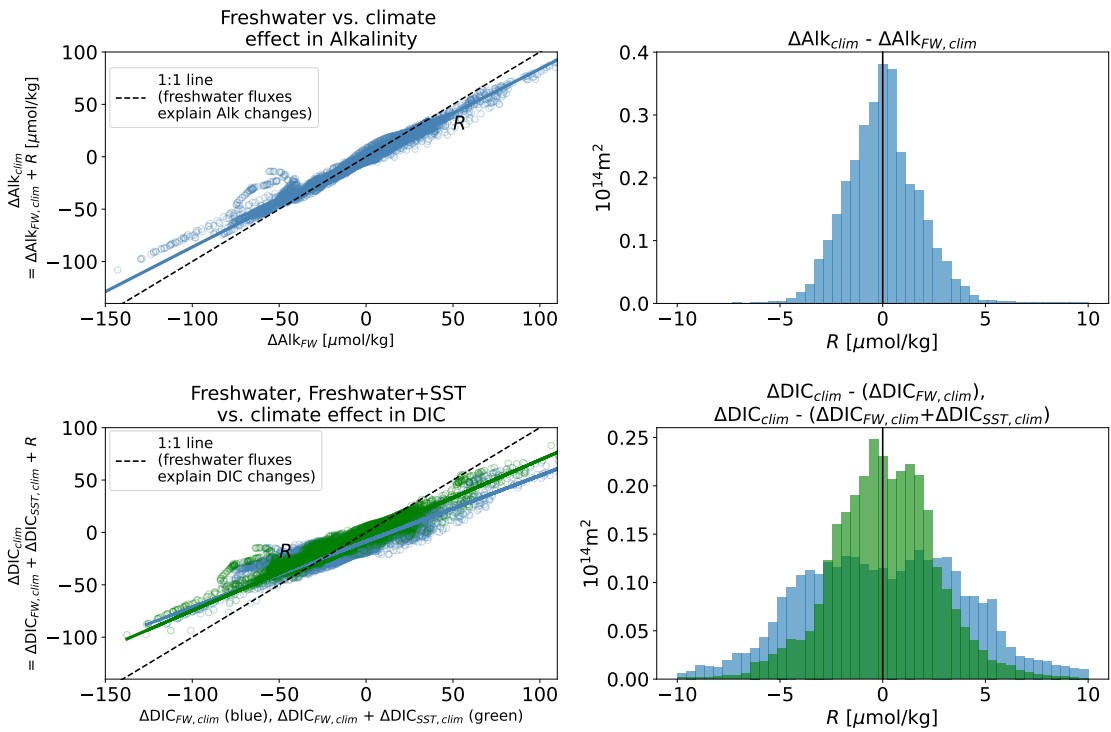

**Figure A2. Ability of freshwater, SST decomposition to reconstruct spatial pattern of changes in surface DIC and Alkalinity with climate effect.** As in Figure A1 but for the climate effect. The greater disagreement here is primarily due to anomalous fluxes.





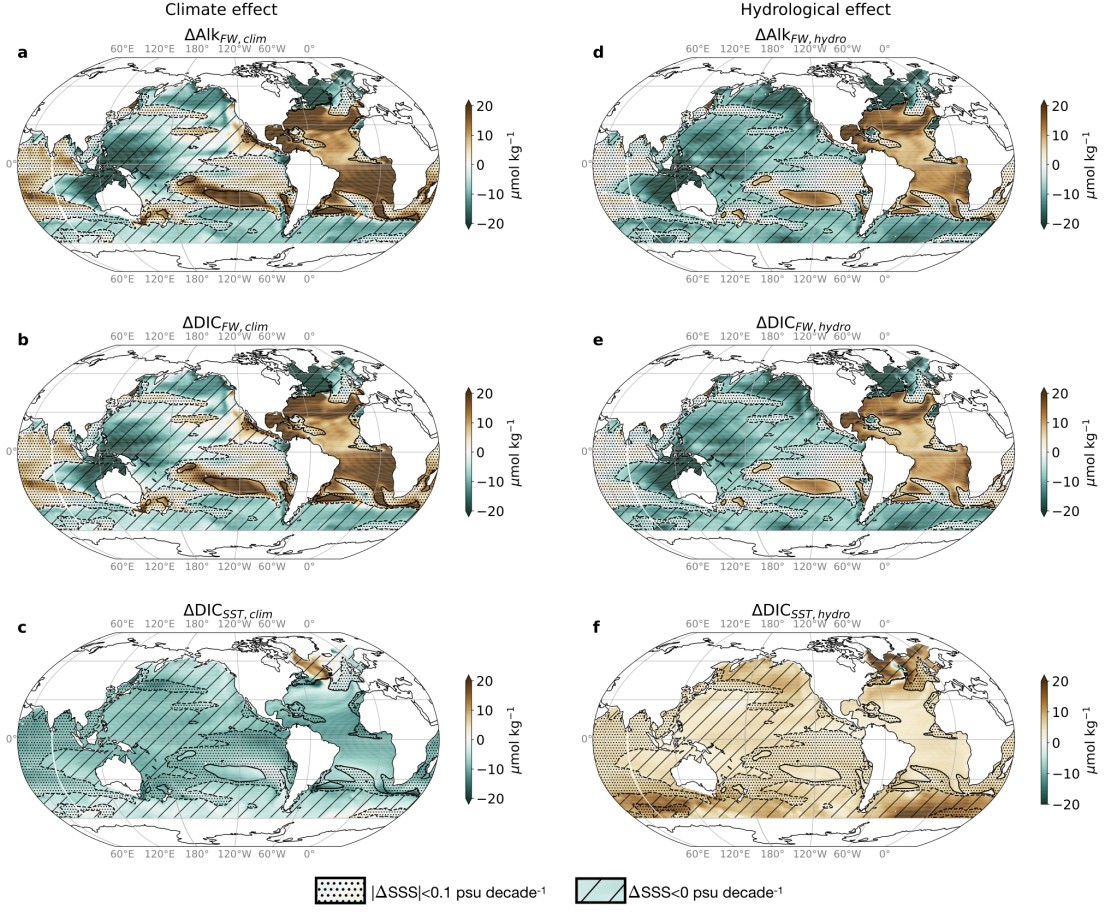

**Figure A3. Thermal and freshwater components of DIC and Alkalinity changes.** (a-c) change attributed to freshwater ($\Delta\mathrm{Alk}_{FW,clim} = f_{FW,clim}\mathrm{Alk}_{Fix-Clim}$, $\Delta\mathrm{DIC}_{FW,clim} = f_{FW,clim}\mathrm{DIC}_{Fix-Clim}$) and thermal ($\Delta\mathrm{DIC}_{thermal,clim}$ = -7$\Delta\mathrm{SST}_{clim}$) components in climate effect; (d-f) change attributed to freshwater ($\Delta\mathrm{Alk}_{FW,hydro} = f_{FW,hydro}\mathrm{Alk}_{Fix-SSS}$, $\Delta\mathrm{DIC}_{FW,hydro} = f_{FW,hydro}\mathrm{DIC}_{Fix-SSS}$) and thermal ($\Delta\mathrm{DIC}_{thermal,hydro}$ = -7$\Delta\mathrm{SST}_{hydro}$) components in hydrological effect.



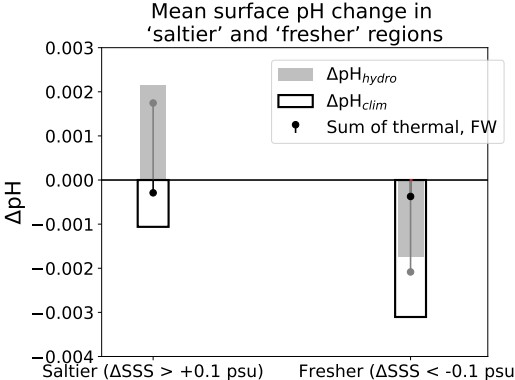

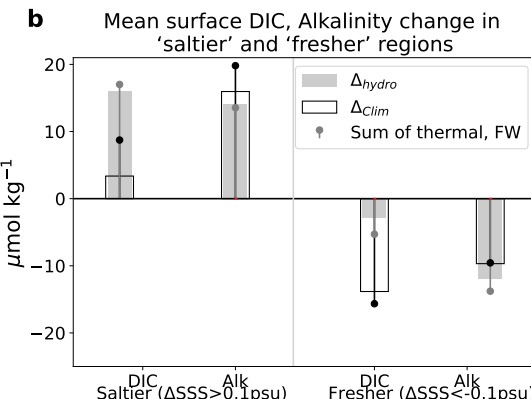

**Figure A4.** Mean change in 'salty-get-saltier' and 'fresh-get-fresher' regions, as in Figures 2c, f and 4b, e (bars), with comparison to sum of freshwater flux and thermal components. In each case, the empty black bars and black lollipops correspond to the climate effect while the grey filled bars and grey lollipops correspond to the hydrological effect.



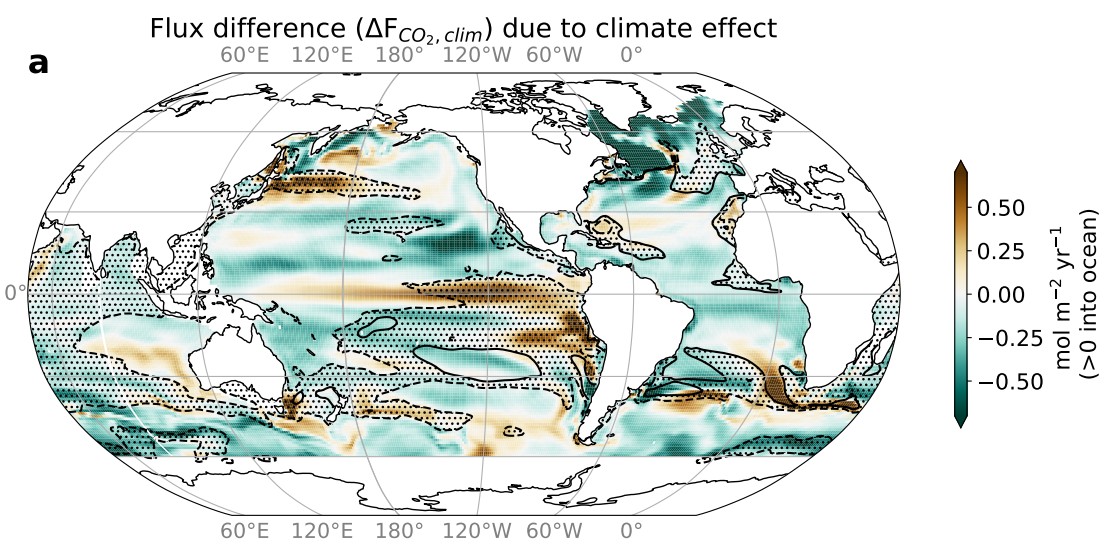

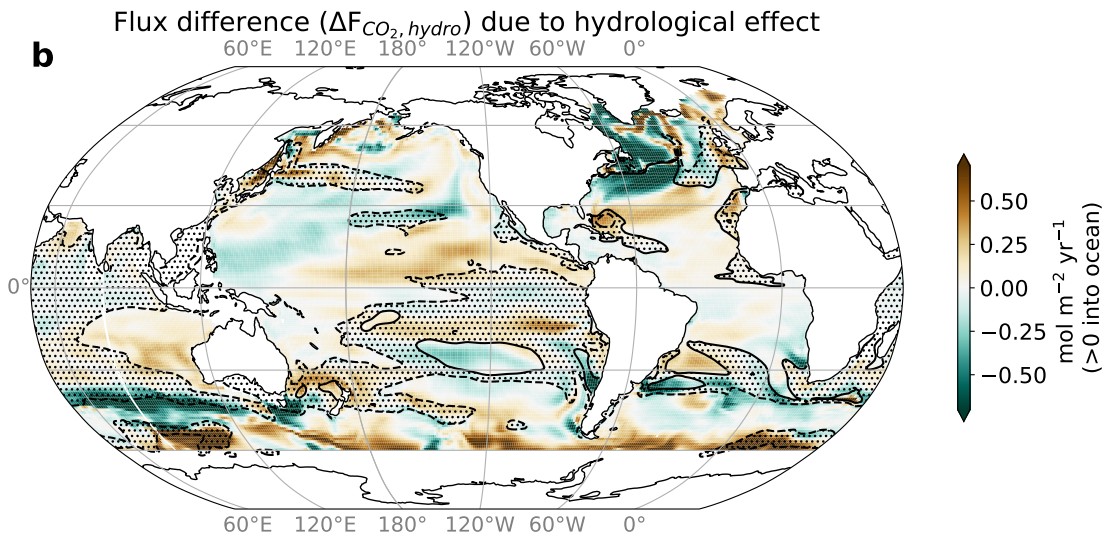

**Figure A5.** Air-sea flux of $CO_2$, difference due to (a) climate effect and (b) hydrological effect. Stippling where $\Delta SSS_{hydro}$ is small, as in Figures 1, 2, 4, 5.



*Author contributions.* A.H. contributed to design and execution of experiments, data analysis, and writing. L.R. contributed to design of
experiments and writing.

*Competing interests.* The authors declare no competing interests.

*Acknowledgements.* A.H. acknowledges support from the National Science Foundation Graduate Research Fellowship Program under Grant
No. DGE-2039656. Any opinions, findings, and conclusions or recommendations expressed in this material are those of the author and do
not necessarily reflect the views of the National Science Foundation. The authors thank the Princeton Institute for Computational Science
and Engineering (PICSciE) for High Performance Computing (HPC) provision, storage and support.





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
