# Peer review of "Hydrological cycle amplification imposes spatial pattern on climate change response of ocean pH and carbonate chemistry"

_EGUsphere, 2024_

## Author Response (AR1)

Please find below the text of both reviews, and our responses to each query in italics. We thank both reviewers for their careful examination of the manuscript and helpful suggestions. In response to concerns about clarity, we have

   (a) Added three tables which describe the model simulations, decomposition of DIC and Alkalinity, and decomposition of pH. This excellent suggestion from reviewer #1 will help future readers follow the abbreviations we use throughout the text. Tables are found in this response below Reviewer 1's comment #7.
   (b) Modified the text throughout to clarify the methods in our study and the relationship of our findings to prior work in the field. Notable changes to the text are in response to Reviewer 1's comments 3, 4, 8. Important Methods points are clarified in response to Reviewer 2's comments 3, 5 and Appendix figures updated in response to Reviewer 2's comments 2, 10.

Sincerely,
Dr. Allison Hogikyan, Dr. Laure Resplandy

—--------------------------------------------
Reviewer 1.

This paper assesses the impacts of the hydrological cycle on pH changes due to increasing atmospheric CO2. It focuses on the dilution and concentration effects of dissolved inorganic carbon (DIC) and alkalinity at the ocean surface, and it also considers the cooling effect of an amplified hydrological cycle. These processes are emphasized as key drivers of ocean surface pH trends, a finding that is novel and important, warranting publication.

   1. However, the overall structure of the paper is not well-organized, making it challenging to understand the concepts presented. I recommend a major revision to enhance clarity and readability for the audience.

*To enhance clarity, we made a table with definitions of model simulations, decomposition of DIC and Alkalinity, and decomposition of pH, as you suggested in your minor point #7 (see details below). We also clarified the text, as detailed below. Briefly, this includes clarification in methods (points #2, 3), as well as in introduction and discussion (points #4 and 6). We also expanded the description of Figure 1 a,b presenting the cooling effect of hydrological amplification (points #8) and moved the figure reference from the introduction to the results section.*

   2. The authors argue that the hydrological effects primarily dilute the negative ions that buffer CO2 dissolution (as stated in the abstract), leading to acidification in

> regions where salinity becomes lower. However, it is unclear why positive ions, like H+, are not also diluted in these regions. I may have confused this point.

*You're correct that all ions would be diluted. We take into account the dilution/concentration of both negative and positive ions in our approach. We estimate the net change in pH from change in Alkalinity (defined as the net change in charge balance).*

*This is done by estimating the pH that would result from dilution/concentration of DIC and Alkalinity with CO2SYS (this example for the hydrological effect):*

*$pH(DIC_{Fix\text{-}SSS} + f_{FW}*DIC_{Fix\text{-}SSS}, Alk_{Fix\text{-}SSS} + f_{FW}*Alk_{Fix\text{-}SSS}, SST_{Fix\text{-}SSS}, SSS_{Fix\text{-}SSS})$*

*And comparing it to the pH without dilution/concentration of DIC, Alk, also calculated by CO2SYS for consistency:*

*$pH(DIC_{Fix\text{-}SSS}, Alk_{Fix\text{-}SSS}, SST_{Fix\text{-}SSS}, SSS_{Fix\text{-}SSS})$*

*The difference between these two quantities gives an estimate of the change in pH due to dilution/concentration of DIC, Alk.*

*This definition is paraphrased from lines 145-152. The freshwater component of the climate effect would be referenced to the Fix-Clim experiment everywhere that the above is referenced to the Fix-SSS experiment. These definitions are also found in the tables we added to the new manuscript (see point #7).*

3. Aren't the changes in vertical transport due to mixing changes (like w'dC/dz vs w'dA/dz) important? I assumed that salinity-enhanced mixing in these regions would increase surface DIC and alkalinity from deeper, richer waters. However, due to the larger vertical gradient in alkalinity compared to the DIC gradient—since surface water accumulates anthropogenic carbon—this effect should be more pronounced in alkalinity than DIC. Is this effect not important?

*This point was indeed unclear as also pointed out by reviewer #2. We have clarified the limitation of estimating mixing and transport in the method we use (i.e. mixing/transport of DIC and Alk are included in f_FW and inferred from salinity) and how we minimize this effect by restricting the analysis to the mixed layer. We modified the Methods text to read: "The effects of mixing and transport on salinity are included in f_FW, because it is derived from changes in salinity. However, since mixing and transport act on different spatial gradients for each variable, f_FW cannot be expected to be the same for salinity, DIC, Alk, etc., except in the mixed layer where gradients are relatively weak for all constituents. As a consequence, we restrict our analysis to the mixed layer, where this error is small."*

4. Most carbon chemistry research decomposing thermal and salinity effects on DIC and alkalinity have analyzed salinity-normalized DIC and alkalinity, which extracts salinity-driven biogeochemical variability beyond dilution and concentration effects. This paper challenges that approach by using a salinity-constrained experiment instead. The differences between these analyses remain unclear. What is the strong point of this analysis? (It is mentioned in Section 2.2, but not sufficient.) From your analysis, what biases might we be introducing into the salinity normalized analysis referenced to 35-psu? Conversely, what is the bias in your surface salinity-constrained experiment method? More discussion on this would greatly benefit future research.

*Thanks for bringing this up. We have clarified the final paragraph of discussion to expand on this point and clarify the questions posed by the reviewer as follows:*
*" Finally, it is worthwhile to note that our 'freshwater' component is similar to the traditional salinity normalization, in that both make use of the fact that freshwater fluxes should change DIC and Alkalinity approximately in proportion to salinity. However, we reference a spatially resolved pre-industrial control salinity to describe the effect of climate change rather than referencing the central estimates of 1900 umol/kg DIC, 2310 umol/kg Alk, and 35 psu used in the standard normalization (Broecker and Peng 1995). Our method makes fewer assumptions and is slightly more precise, but requires more data.*

*Our study also demonstrates that this 'freshwater effect' (which the salinity normalization is intended to remove) can have substantial consequences for carbonate chemistry, with implications for CO2 fluxes, pH, etc. This result raises the question of how useful salinity-normalized values are. If one only examined the salinity-normalized CO2, for example, one might struggle to explain changes in other carbonate system parameters, especially in a scenario with strong freshwater fluxes."*

Minor Revisions:

5. Please refer to papers suggesting CO2-concentration feedback. The term "this study" in line 20 is unclear without the citation.

*Done.*

6. L56: "These changes in salinity modify ocean circulation and lead to enhanced ocean heat uptake" remains unclear. I believe that an increase in salinity enhances heat uptake but not a decrease. Moreover, why does an increase in salinity enhance heat uptake? Is it because enhanced mixing reduces the surface temperature? Since this concept is key for this paper, please describe

this in more detail. The reference for this description in L58 should be Figure 1d, not Figure 1b.

*It's true that a decrease in salinity can locally decrease OHU. However, the net effect of hydrological cycle amplification in the global mean in climate models is to enhance OHU, as shown by prior studies. We clarified in introduction as follows: "these changes in salinity modify the ocean circulation and lead to a net increase in ocean heat uptake globally, which weakens surface warming (as shown by Liu et al., 2021; Williams et al. 2007). This additional heat uptake is attributed  to enhanced subduction in regions of strong sea surface salinity increase, primarily in the North Atlantic."*

7. The paper includes a variety of simulations and decomposition analyses to isolate climate and hydrological effects. The jargon used for simulation names often causes confusion; therefore, I recommend creating a table listing all experiments, their factor decomposition, and short descriptions.

*This is a nice idea, and we have inserted three tables in the Methods section- attached below- describing the three model experiments, the decompositions for DIC and Alk, and the decomposition for pH. The tables are then referenced in the main text where we introduce the model simulations, DIC/Alk decomposition, and pH decomposition, respectively, and in the caption of each figure.*

*After a new paragraph at the beginning of Results (see your query #8): "Please see Table 1 and the Methods section for an overview of the experiments we use to isolate the climate effect and hydrological effect."*

*At line 161 in the old text: "Where SSS increases, DIC is less sensitive than Alk to the climate effect ( DDICclim = +2 umol/kg and DAlkclim = +16 umol/kg; see Table 2 for definitions)"*

*At line 215 in the old text: "The climate effect tends to decrease surface pH, thereby re-inforcing the acidification associated with the rise in atmospheric CO2 (DpHclim <0; Figure 4a; see Methods and Table 3} for definition of all DpH terms)."*

**Table 1.** Simulation definitions

| Standard | $CO_2$ increases at 1% per year from 286 to 572 ppm (requires 70 years) then is held at 572 ppm for another 30 years, for 100 total simulation years |
|---|---|
| Fix-SSS | $CO_2$ trajectory is as in the Standard experiment, and SSS is restored to pre-industrial concentrations |
| Fix-Clim | Model is heated following the $CO_2$ trajectory of the Standard experiment, but additional $CO_2$ does not interact with model chemistry (e.g. there is no ocean carbon uptake) |

**Table 2.** $\Delta$DIC, $\Delta$Alk component definitions. $X$ stands for DIC and Alk.

| | | |
|---|---|---|
| $\Delta X_{Hydro}$ | Change in X due to hydrological cycle amplification (averaged over simulation years 71-100) | $\Delta X_{Hydro} = X_{Standard} - X_{Fix-SSS}$
$\Delta X_{Hydro} = \Delta X_{FW,hydro} + \Delta X_{therm,hydro} +$ residual$_{hydro}$ |
| $\Delta X_{Clim}$ | Change in X due to climate effect (averaged over simulation years 71-100) | $\Delta X_{Clim} = X_{Standard} - X_{Fix-Clim}$
$\Delta X_{Clim} = \Delta X_{FW,clim} + \Delta X_{therm,clim} +$ residual$_{clim}$ |
| $\Delta X_{FW,Hydro}$ | Change in X due to dilution/concentration from hydrological effect | $\Delta X_{FW} = f_{FW} \cdot X_{Fix-SSS}$
where $f_{FW} = \frac{SSS_{Standard} - SSS_{Fix-SSS}}{SSS_{Fix-SSS}}$ |
| $\Delta X_{FW,Clim}$ | Change in X due to dilution/concentration from climate effect | $\Delta X_{FW} = f_{FW} \cdot X_{Fix-Clim}$
where $f_{FW} = \frac{SSS_{Standard} - SSS_{Fix-Clim}}{SSS_{Fix-Clim}}$ |
| $\Delta DIC_{thermal,Hydro}$ | Change due to temperature change from hydrological effect, undefined for Alk | $-7\frac{\mu mol/kg}{K} \cdot (SST_{Standard} - SST_{Fix-SSS})$ |
| $\Delta DIC_{thermal,Clim}$ | Change due to temperature change from climate effect, undefined for Alk | $-7\frac{\mu mol/kg}{K} \cdot (SST_{Standard} - SST_{Fix-Clim})$ |

**Table 3.** $\Delta$pH component definitions.

| | | |
|---|---|---|
| $\Delta pH_{Hydro}$ | Change in pH due to hydrological cycle amplification using CO2SYS | CO2SYS-based pH from Standard variables minus CO2SYS-based pH from Fix-SSS variables.
$\Delta pH_{Hydro} = \Delta pH_{FW,Hydro} + \Delta pH_{therm,Hydro} +$ residual$_{Hydro}$ |
| $\Delta pH_{Clim}$ | Change in pH due to climate effect using CO2SYS | CO2SYS-based pH from Standard variables minus CO2SYS-based pH from Fix-Clim variables.
$\Delta pH_{Clim} = \Delta pH_{FW,Clim} + \Delta pH_{therm,Clim} +$ residual$_{Clim}$ |
| $\Delta pH_{FW,Hydro}$ | Change in pH due to dilution/concentration from hydrological effect using CO2SYS | CO2SYS-based pH from Fix-SSS variables + $\Delta X_{FW,Hydro}$ $(DIC_{Fix-SSS} + \Delta DIC_{FW,Hydro}$, $Alk_{Fix-SSS} + \Delta Alk_{FW,Hydro}$, $SST_{Fix-SSS}$, $SSS_{Fix-SSS})$ minus CO2SYS-based pH$_{Fix-SSS}$ |
| $\Delta pH_{FW,Clim}$ | Change in pH due to dilution/concentration from climate effect using CO2SYS | CO2SYS-based pH from Fix-Clim + $\Delta X_{FW,Clim}$ $(DIC_{Fix-Clim} + \Delta DIC_{FW,Clim}$, $Alk_{Fix-Clim} + \Delta Alk_{FW,Clim}$, $SST_{Fix-Clim}$, $SSS_{Fix-Clim})$ minus CO2SYS-based pH$_{Fix-Clim}$ |
| $\Delta pH_{thermal,Hydro}$ | Change in pH due to temperature change from hydrological effect using CO2SYS | CO2SYS-based pH from Fix-SSS variables + $\Delta DIC_{thermal,Hydro}$ $(DIC_{Fix-SSS} + \Delta DIC_{thermal,Hydro}$, $Alk_{Fix-SSS}$, $SST_{Standard}$, $SSS_{Fix-SSS})$ minus CO2SYS-based pH$_{Fix-SSS}$ |
| $\Delta pH_{thermal,Clim}$ | Change in pH due to temperature change from climate effect using CO2SYS | CO2SYS-based pH from Fix-Clim variables + $\Delta DIC_{thermal,Clim}$ $(DIC_{Fix-Clim} + \Delta DIC_{thermal,Clim}$, $Alk_{Fix-Clim}$, $SST_{Standard}$, $SSS_{Fix-Clim})$ minus CO2SYS-based pH$_{Fix-Clim}$ |

8. Figure 1 is introduced in the Introduction and contains simulation names before their descriptions. While it underscores the importance of the cooling effect

through the amplified hydrological cycle, its early reference in the Introduction is unfriendly to readers. If used in the Introduction, a more detailed description of the experiments and isolated effects is needed.

*We removed the reference to Figure 1 from the introduction to avoid the issue raised by the reviewer and limit the Introduction to reference prior studies. We now present Figure 1a,b and the cooling effect at the beginning of the results section as follows: :*

*"The climate model used here, ESM2M, responds to a CO2 increase with a surface warming and enhancement of mean salinity patterns. This results in a salinity increase in salty subtropical regions (enhanced in the Atlantic, relative to the Pacific), and a decrease in fresh regions, most notably high latitudes and the western tropical Pacific Ocean (Figure 1a, b). These changes represent the 'climate effect' in SST and SSS, and are consistent with many prior studies (most notably Manabe and Wetherald 1975, Held and Soden 2006), and are similar to changes in both SST and SSS seen in historical trends (Figure 1e, f; see Durack et al. 2010). While this 'climate effect' has been studied, and hydrological cycle amplification is known to be a robust feature of global warming, the effect of hydrological cycle amplification, or 'hydrological effect', has only been isolated more recently (Williams et al. 2007, Liu et al. 2021, Hogikyan et al. 2023). As shown in these prior studies, the hydrological effect accounts almost exactly for the SSS changes in the climate effect and leads to a surface cooling (due to enhanced global ocean heat uptake) (Figure 1c, d)."*

9. Line 108: The statement "Alkalinity is not directly sensitive to temperature" is confusing. Consider referencing Figure 1 to suggest the cooling effect, discussing solubility changes before this description, and noting that alkalinity does not change even with surface solubility changes.

*We propose to clarify this statement by simplifying it: "Alkalinity does not vary with temperature."*

10. Appendix figures should be referenced in the main text. It is unclear which part each appendix figure corresponds to.

*Appendix figures are referenced in the main text, with an 'A' before the figure number. At line 110 where this notation first appears, we added: "Note AX indicates Appendix Figure X"*
* * *
Reviewer 2.

This study uses experiments with an Earth system model to investigate the effects of changes in the hydrological cycle on surface DIC, alkalinity, and pH. The methodology employed is a relatively simple linear approach, yet robust, allowing for the separation of the direct effect of hydrological changes from the indirect effect of the accompanying temperature changes. Highlights and novel insight include that while the effect of changes in hydrology is small in terms of global pH, it plays a significant role in driving the regional variability in pH due to climate change. Specifically, the effect of changes in hydrology opposes acidification in regions that become "saltier", but enhances acidification in regions that will become fresher (hence the small overall effect on global scale).

In my opinion the study is novel, interesting and provides much needed insight for the drivers of the spatial variability in the response of the DIC, Alkalinity and pH to climate change. In my opinion, the methodology and analysis is thorough. While some necessary assumptions are made, the study provides useful information regarding the first-order effects of changes in the hydrological cycle. I recommend the study to be accepted for publication after addressing some minor queries/suggestions/comments detailed below (mostly in terms on clarifications to further strengthen the study).

General comment:
   1.    I appreciate that assumptions are necessary for separating the different drivers in terms of changes in pH. However, **I** think it will be useful to be able to see how robust is your assumptions in section 2.3. I suggest you include a figure (maybe in the appendix) that shows maps of the actual DeltapH_hydro (pH_standard – pH_fix-sss) and DeltapH_clim (pH_standard – pH_fix-clim) from the model experiments versus the DeltapH_hydro and DeltapH_clim estimated from pyCOSYS.

*We initially chose not to include these maps because pH has a highly nonlinear dependence on DIC, Alk, and temperature; the residual at a given location (on the map you ask about) is large. We get an improved signal to noise ratio, and therefore learn more, by aggregating over 'freshening' and 'salinifying' regions, which is the result shown in Figures 4 and 5. We also provide an estimate of the error, aggregated over these regions, in Figure A4- the reviewer pointed out that the caption was not clear for this figure and we have clarified it (query #10). However, if requested by reviewers, we will add the maps of the 'predicted' pH from the linear approximations in comparison to the actual simulated pH to the Appendix.*

Specific comments:
   2.    Lines 111-113. I am not sure I agree with this. Based on comparisons of the panels in Figure A5 to me it appears that the hydrological cycle leads to equally large "anomalous CO2 fluxes" as the climate effect. In my opinion, this makes

sense as in my understanding a large part of the effect from changes in the circulation is encapsulated to this "hydrology cycle". Please, clarify and correct me if I have misunderstood.

*We agree that it is hard to see the magnitude of the net flux in the maps; we added integrated estimates as in the main text figures to clarify that this statement is true (see figure below: the empty bars associated with the climate effect are larger than the grey bars associated with the hydrological effect). To your second question: Globally, circulation changes due to warming are stronger than those due to hydrological cycle amplification, although in some regions they are equally strong (if you're interested, there is more information on circulation changes due to the hydrological vs. climate effects in Hogikyan et al. 2023 focusing on ocean oxygen).*

[Figure]

3.       Lines 125-126. "This approach includes the influence of mixing and transport …" This is a little misleading, as I agree with you in terms of what you describe/discuss in lines 115 and 127-129 that this approach does not include and cannot account for the influence of mixing and transport as the vertical and horizontal gradients of salinity and DIC and alkalinity are not the same. Please consider erasing this part or rephrase it to avoid confusion.

*Great point, and thanks for your careful inspection. We think it will be helpful for readers to say something on this point and propose to include the following in Methods: "The effects of mixing and transport on salinity are included in f_FW, because it is derived from changes in salinity. However, since mixing and transport act on different spatial gradients for each variable, f_FW cannot be expected to be the same for salinity, DIC,*

*Alk, etc., except in the mixed layer where gradients are relatively weak for all constituents. As a consequence, we restrict our analysis to the mixed layer, where this error is small."*

4.      Section 2.3 and equations. Throughout this section you use different symbols/notation for the runs than in the previous section which makes it a little confusing. I suggest, for consistently, to use in all equations/notation: "Standard" instead of "Std" and "thermal" instead of "T" (such as DeltaDIC_thermal,hydro in line 151).

*Yes, this should be "thermal" everywhere; there was a typo. This is corrected.*

5.      Section 2.3 estimates of DeltapH. In my opinion, it makes more sense to reference the DeltapH in a pH_fix-SSS that has been estimated using a consistent approximation (the CO2SYS algorithm and not the direct pH from the fix-sss and fix-clim runs), which does not seem to be the case looking at the equations in lines 148-153. To clarify, I propose that the pH_fix-sss in the equations in lines 148-153 should be estimated as pH(DIC_fix-sss,Alk_fix-sss,SST_fix-sss,SSS_fix-SSH) from the CO2SYS algorithm rather than using the pH directly from the fix-sss run. Else, in my opinion, you could be introducing additional unrealistic changes in the DeltapH. Please correct me if I have misunderstood.

*We agree and this is in fact what we have done. To clarify, we have substantially modified the text throughout the Methods section to emphasize that the pH attributions are all done with CO2SYS for consistency. The sentence at line 142 is changed to: "... the 'Standard' as well as 'Fix-SSS', 'Fix-Clim' pH are all estimated using the marine carbonate chemistry solver PyCO2SYS (Humphreys et al. 2022) to remove any biases between CO2SYS and the earth system model." In addition, we have added a table that lists the terms in our decomposition and describes how each term is computed (below).*

**Table 3.** $\Delta$pH component definitions.

| | | |
|---|---|---|
| $\Delta\text{pH}_{Hydro}$ | Change in pH due to hydrological cycle amplification using CO2SYS | CO2SYS-based pH from Standard variables minus CO2SYS-based pH from Fix-SSS variables. $\Delta\text{pH}_{Hydro} = \Delta\text{pH}_{FW,Hydro} + \Delta\text{pH}_{therm,Hydro} + \text{residual}_{Hydro}$ |
| $\Delta\text{pH}_{Clim}$ | Change in pH due to climate effect using CO2SYS | CO2SYS-based pH from Standard variables minus CO2SYS-based pH from Fix-Clim variables. $\Delta\text{pH}_{Clim} = \Delta\text{pH}_{FW,Clim} + \Delta\text{pH}_{therm,Clim} + \text{residual}_{Clim}$ |
| $\Delta\text{pH}_{FW,Hydro}$ | Change in pH due to dilution/concentration from hydrological effect using CO2SYS | CO2SYS-based pH from Fix-SSS variables $+ \Delta X_{FW,Hydro}$ $(\text{DIC}_{Fix-SSS} + \Delta\text{DIC}_{FW,Hydro}, \quad \text{Alk}_{Fix-SSS} + \Delta\text{Alk}_{FW,Hydro}, \text{SST}_{Fix-SSS}, \text{SSS}_{Fix-SSS})$ minus CO2SYS-based $\text{pH}_{Fix-SSS}$ |
| $\Delta\text{pH}_{FW,Clim}$ | Change in pH due to dilution/concentration from climate effect using CO2SYS | CO2SYS-based pH from Fix-Clim $+ \Delta X_{FW,Clim}$ $(\text{DIC}_{Fix-Clim} + \Delta\text{DIC}_{FW,Clim}, \quad \text{Alk}_{Fix-Clim} + \Delta\text{Alk}_{FW,Clim}, \quad \text{SST}_{Fix-Clim}, \text{SSS}_{Fix-Clim})$ minus CO2SYS-based $\text{pH}_{Fix-Clim}$ |
| $\Delta\text{pH}_{thermal,Hydro}$ | Change in pH due to temperature change from hydrological effect using CO2SYS | CO2SYS-based pH from Fix-SSS variables $+ \Delta\text{DIC}_{thermal,Hydro}$ $(\text{DIC}_{Fix-SSS} + \Delta\text{DIC}_{thermal,Hydro}, \text{Alk}_{Fix-SSS}, \text{SST}_{Standard}, \text{SSS}_{Fix-SSS})$ minus CO2SYS-based $\text{pH}_{Fix-SSS}$ |
| $\Delta\text{pH}_{thermal,Clim}$ | Change in pH due to temperature change from climate effect using CO2SYS | CO2SYS-based pH from Fix-Clim variables $+ \Delta\text{DIC}_{thermal,Clim}$ $(\text{DIC}_{Fix-Clim} + \Delta\text{DIC}_{thermal,Clim}, \text{Alk}_{Fix-Clim}, \text{SST}_{Standard}, \text{SSS}_{Fix-Clim})$ minus CO2SYS-based $\text{pH}_{Fix-Clim}$ |

6.        Lines 141-142. I am not sure what you mean by "… ESM2M does not reach chemical equilibrium because it is constrained by conservation of heat and mass". Maybe I have misunderstood but just to clarify I believe that your assumption is that the surface ocean is in a saturated state? In my understanding this does not have to be the case not just because the model is an ESM that conserves heat and mass. Please, consider clarifying.

*Thanks for noting that this is unclear, this sentence is meant to explain exactly what you point out above in your query #5. We clarified the text: "These DpH_Clim and DpH_Hydro estimates from CO2SYS are not identical to pH_Std - pH_Fix-Clim and pH_Std - pH_Fix-Hydro from the model experiments because CO2SYS assumes chemical equilibrium. (ESM2M is constrained by the conservation of heat and mass but a given location is not necessarily in chemical equilibrium)."*

7.        Lines 165-166. I believe that your residual includes not only biological or circulation effects but also the effect of the non-linearity (i.e., errors associated with the linear assumption of your methodology). I suggest that you should explicitly discuss/mention the inclusion of this non-linear effect within the residual here to avoid any confusion.

*Yes, we have expanded the phrase here to include this explicitly: "residual (e.g. approximations in f_FW, as well as biological and circulation)"*

8. Line 200-201 (links to my comment above), I believe that the residual will also include the effect of the nonlinearity of the carbonate system. I suggest you add this in the text here.

*We made this more specific: "Major processes that are not included in these freshwater and thermal effects include air-sea $CO\_2$ fluxes and other adjustments of the carbonate system, spatial shifts in the atmospheric and oceanic circulations, and biological activity. Errors in our method are also included in the residual."*

9. Line 211-212. I am not sure what you mean by the "efficiency of the 3D mixing", maybe consider rephrasing to something along the lines "… the effect of vertical and horizontal mixing, and circulation on the near-surface ocean…".

*We propose to simplify the phrasing: "the DIC change due to a given surface flux is sensitive to multiple factors, including the effects of mixing and advection, as well as the temperature, surface wind speed, and sea state."*

10. Figure A4. I am a little confused by this figure. (i) I suggest to clarify in the caption-text what the bars correspond to (I think is the freshwater component/effect but I may have misunderstood). (ii) I think there must be a typo here as you reference Figure 4b and e but figure 4 has no panel e, maybe you mean c (please check). (iii) In the legend you state that the "lollipops" correspond to the sum of thermal and freshwater. I believe you probably mean the thermal + freshwayer + residual, else I am unsure how you separated the thermal contribution in the alkalinity from the residual (and in my understanding this residual in alkalinity includes the effect of circulation, mixing, nonlinearity etc. rather than only the thermal contribution). Please consider clarifying in the caption and update the legend.

*Yes, there is no thermal component for Alk. This is now noted in the new Table 2 with definitions of variables in our decomposition for DIC and Alkalinity, and more clearly stated in Methods: "Alkalinity does not vary with temperature".*

*We also greatly expanded the caption : "Error in our estimation of pH, DIC, Alk changes, averaged over `salty-get-saltier' and `fresh-get-fresher' regions. Bars of pH change in (a) are identical to those in Figure 4c, while bars of DIC, Alk change in (b) are identical to those in Figure 2e (i.e. empty black bars correspond to the climate effect while the grey filled bars correspond to the hydrological effect). Lollipops make a comparison between the estimate from our decomposition, and the full changes simulated by ESM2M (represented by the bars). Grey lollipops represent $DX_{FW, Hydro} + DX_{thermal, Hydro}$ (X is pH, DIC, or Alk), and the difference between the grey bars and lollipops is $R_{Hydro}$. Black lollipops represent $DX_{FW, Clim} + DX_{thermal, Clim}$ (X is pH, DIC, or Alk), and the difference between the black bars and lollipops is $R_{Clim}$. Please see Methods and Tables 2, 3 for calculations of FW, thermal components."*

Typos:

Line 108. I believe you mean Figures A1 and A2 (not S1 and S2).

*Absolutely.*

Lines 174-175, decrease appears twice (rephrase to something along the lines "…
decrease DIC and Alkalinity by -9 …").

*Done, thanks.*

Line 185-186. I believe you mean Figure A3 c and f.

*Yes.*

Figures 4 and 6 caption-text. I believe you mean "… hatching indicates DSSS_hydro<
0.1" not <0 (based on the rest of the discussion and the actual legends in the plots).
Please check.

*Yes, thanks.*

Figure A3, for consistency with the rest of the text and the equations, I suggest you
change the title in c and f to DeltaDIC_thermal,clim and DeltaDIC_thermal,hydro (rather
than _SST

*Done.*

---

## Author Response (AR2)

Reply to Associate Editor Manmohan Sarin regarding "Hydrological cycle amplification imposes spatial pattern on climate change response of ocean pH and carbonate chemistry" by Allison Hogikyan and Laure Resplandy.

1. Abstract, line 6: Authors may like to replace the word 'culprit' with mechanism or process.

The sentence now reads: "Here, we investigate spatial patterns in the climate effect on surface-ocean acidification (and the closely related carbonate chemistry) in an Earth System Model under a rapid CO2-increase scenario, and identify a different driving process."

2. Section 2.2, line 127: Please consider clarifying 'where this error is small'. This sounds very qualitative statement!

The sentence now reads: "...we restrict our analysis to the mixed layer, where this error is very small relative to the changes driven by temperature and freshwater effects (as demonstrated in Figures 3 and A1, A2).

Thank you for your careful consideration.

Sincerely,

Dr. Hogikyan, Dr. Resplandy